# Characterization of Spatial Air Pollution Patterns Near a Large Railyard Area in Atlanta, Georgia

**DOI:** 10.3390/ijerph16040535

**Published:** 2019-02-13

**Authors:** Halley L. Brantley, Gayle S.W. Hagler, Scott C. Herndon, Paola Massoli, Michael H. Bergin, Armistead G. Russell

**Affiliations:** 1Department of Statistics, North Carolina State University, Raleigh, NC 27607, USA; hlbrantl@ncsu.edu; 2U.S. EPA Office of Research and Development, Research Triangle Park, NC 27711, USA; 3Aerodyne Research Inc., Billerica, MA 01821, USA; herndon@aerodyne.com (S.C.H.); pmassoli@aerodyne.com (P.M.); 4Department of Civil and Environmental Engineering, Duke University, Durham, NC 27708, USA; michael.bergin@duke.edu; 5Civil and Environmental Engineering, Georgia Institute of Technology, Atlanta, GA 30332, USA; ted.russell@ce.gatech.edu

**Keywords:** near-source, railyard, air pollution, mobile monitoring, locomotive

## Abstract

Railyards are important transportation hubs, and they are often situated near populated areas with high co-located density of manufacturing, freight movement and commercial enterprises. Emissions occurring within railyards can affect nearby air quality. To better understand the air pollution levels in proximity to a major railyard, an intensive mobile air monitoring study was conducted in May 2012 around a major railyard area in Atlanta, GA, constituted of two separate facilities situated side-by-side. A total of 19 multi-hour mobile monitoring sessions took place over different times of day, days of the week, and under a variety of wind conditions. High time resolution measurements included black carbon (BC), particle number concentration (PN), particle optical extinction (EXT), oxides of nitrogen (NO, NO_2_, NOy), carbon monoxide (CO), and speciated air toxics. Urban background was estimated to contribute substantially (>70%) to EXT and CO, whereas BC, oxides of nitrogen (NOx) and toluene had comparably low background contributions (<30%). Mobile monitoring data were aggregated into 50 meter spatial medians by wind categories, with categories including low speed wind conditions (<0.5 m s^−1^) and, for wind speeds above that threshold, by wind direction relative to the railyard. Spatial medians of different pollutants measured had a wide range of correlation—gas-phase air toxics (benzene, toluene, acetaldehyde) had moderate correlation with each other (r = 0.46–0.59) and between toluene and CO (r = 0.53), but lower correlation for other pairings. PN had highest correlation with oxides of nitrogen (r = 0.55–0.66), followed by BC (r = 0.4), and lower correlation with other pollutants. Multivariate regression analysis on the full set of 50 m medians found BC and NO as having the strongest relationship to railyard emissions, in comparison to their respective background levels. This was indicated by an increase associated with transiting through the yard and inverse relationship with distance from the railyard; NO and BC decreased by a factor of approximately 0.5 and 0.7 over 1 km distance of the railyard boundary, respectively. Low speed, variable wind conditions were related to higher concentrations of all measured parameters.

## 1. Introduction

Characterization of air quality patterns in near-source environments is a topic of ongoing interest, as studies demonstrate adverse health conditions associated with proximity to transportation sources [1]. Within the United States, it is estimated that 45 million Americans reside in close proximity to large transportation sources, including four-lane roadways, railroads, and airports [2]. Over the last decade, a substantial number of field studies have been conducted to evaluate air pollution trends in near-road environments, finding repeatable exponential increases in concentrations of directly emitted pollutants (e.g., carbon monoxide, nitrogen oxide, black carbon) with proximity to a major highway [3]. Additionally, near-road studies have also revealed significant variability in local air pollution concentrations related to the surrounding topography (barriers, buildings, roadway design), traffic patterns, and local meteorology [4,5]. 

Local-scale air pollution related to other transportation source types—railyards, ports, and airports—has also been characterized in field studies, albeit to a lesser extent than highway settings. These types of sources are complex, with multiple types of activities and related emissions concentrated over a large area. In addition to general concerns about public health in near-source environments, environmental justice concerns are indicated in some locations. Arunachalam, et al. [6] studied several overall goods transportation areas related to a number of major ports, including railyards and major highways, finding higher representation of lower income and minority populations living within 300 m of a source at four of the five port areas studied. 

In the case of railyards, the focus of this study, each railyard is unique in its operations, layout, and geographic setting. Emission sources in classification railyards include switcher locomotives, which are often old model locomotives characterized by high emissions, as well as trains passing through the yard. Intermodal railyards include additional cranes and truck traffic associated with the movement of containers to and from trains. Other possible sources of emissions include service vehicles, small engines, and load-testing of locomotives as part of maintenance.

A recent fine-scale modeling study of a generic intermodal railyard with surrounding urban environment predicted a significant air pollution gradient in downwind locations, with dispersion and the resulting pollutant concentrations strongly modulated by the topography and wind direction [7]. Other dispersion modeling studies have been conducted to evaluate whether railyard emissions may impact local regulatory air monitoring of PM_2.5_, and predicted an increase in PM_2.5_ associated with railyard emissions at the Rougemere Rail Yard in Dearborn, MI [8] and at the Tilford/Inman Rail Yard in Atlanta, GA [9].

Common measurement approaches for near-source monitoring have included stationary and mobile monitoring systems. Stationary field studies have found that isolating local air pollution associated with railyard operations from industrial sources located in close proximity can be difficult [8]. In other studies, near-field elevated concentrations of black carbon [9,10,11], particulate matter [9], and nitric oxide (NO) [11] have been attributed to railyard emissions. Near the Roseville Rail Yard, which included a major locomotive maintenance operation, Cahill et al. [11] also reported the presence of polycyclic aromatic hydrocarbons (PAHs) in the ultrafine (D_p_ < 0.1 µm) fraction of particulate matter. Mobile monitoring, which is the approach used in this present study, provides the ability to conduct observations over a large spatial area and to correlate emission measurements with sources. Therefore, mobile monitoring is becoming an increasingly common strategy of studying air quality and emission sources (e.g., [12,13,14]); however, this approach of measurement is nontrivial in its labor-intensive method of collecting data, and requires both high time-resolution measurement approaches and advanced data analysis methods to process and interpret the results [15]. 

The present study used a mobile monitoring approach to detect spatial patterns of an extensive set of air pollutants surrounding two co-located major railyards in Atlanta, GA. In addition to providing some key findings about air quality proximate the railyard area, these results also provide a useful base case for a railyard area that is expected to eventually replace its older switcher locomotives for new genset locomotives with substantially lower emissions. Finally, the measurement and data analysis methodologies utilized in this study may be replicated in other environments to characterize air quality proximate to large transportation sources. 

## 2. Methods

### 2.1. Field Study Design

A mobile air monitoring study was designed to measure an array of air pollution parameters near a major railyard area in Atlanta, GA. The railyard area includes the Inman and Tilford co-located railyards, owned by CSX and Norfolk Southern, respectively. At the time of the study, the Inman Yard was a large intermodal railyard with 14 switcher locomotives, while the Tilford Yard was a CSX hump classification terminal handling approximately 80 trains per week and operating 10 switcher locomotives [9]. 

The sampling study occurred over the month of May, 2012. The mobile monitoring vehicle was driven for several hour periods (a “session”) on a predesigned route that included areas proximate (<300 m) to the two railyards and on a public roadway that traveled directly through the two yards. The sessions ranged in length from 1.2 to 5.9 h with a mean of 3.8. The sessions consisted of a combination of mobile and stationary sampling with 74% of the data collected while the vehicle was in motion. The mean and median speeds for when the vehicle was in motion were both 12 mph with a standard deviation of 7.8. The data collected during the campaign were divided into 50 m segments and classified into three categories: eastern side, western side, or within railyard (Figure 1). 

A total of 19 sessions, translating to over 70 h of sampling, were conducted and staggered over different timeframes to account for the variability in meteorology and railyard operations. The resulting data cover different hours of day, days of week (with both daytime and nighttime measurements), and a variety of meteorological conditions (Figure 2). Meteorological measurements (1 min) were obtained from a weather station located on the Georgia Institute of Technology (GA Tech) campus (~3 km distant) and were compared with hourly measurements collected at Fire Station 8 directly adjacent to the railyard and at the Atlanta International Airport [9]. Further data analysis used the meteorological measurements at the GA Tech campus because they were available for the entire field campaign. Wind measurements were aggregated into hourly measurements and classified into five categories based on the orientation to the railyard and wind speed: wind speed less than or equal to 0.5 m/s; cross-railyard wind categories of north-northeast (NNE), direction between 337.5 and 112.5 and south-southwest (SSW) direction between 157.5 and 292.5; as well as parallel-to-railyard wind categories including southeast (SE) direction between 112.5 and 157.5; and northwest (NW) between 292.5–337.5. The prevailing wind direction was SE, but winds perpendicular to the primary railyard axis, from the south-southwest (SSW) and north-northeast (NNE), were also observed during the campaign (Figure 2). 

### 2.2. Measurement Approach

The Aerodyne Mobile Laboratory (AML) [16], included an extensive array of instruments (Table 1) measuring carbon monoxide (CO), oxides of nitrogen (NO, NO_2_, NOy), particle number (PN), black carbon (BC), particle optical extinction (EXT), benzene, toluene, and acetaldehyde. All instruments collected data on a real-time basis, with measurement rates ranging from ~1–3 s.

The mobile monitoring instrumentation was connected to a common inlet system. The PM inlet was made of 1” stainless steel tubing which then splits into various copper tubing to travel to different PM instruments, using conductive material in order to prevent excessive particulate wall loss. Flow to smaller PM instruments are split though an isokinetic inlet, with care taken to avoid sharp bends in PM inlet lines. Gas phase inlet lines consist of PFA teflon tubing to minimize any surface chemistry that might occur to sampled species on the walls of the tubing. The gas phase inlet was filtered in order to protect instruments and detectors from particles.

Quality checks were conducted on all instrumentation, including zero and span checks on gas-phase instruments, zero checks via filtering for the particle-phase instruments, and flow checks. Periodically, air from two different zero-air generation sources were used to conduct zero baseline adjustments for instrumentation, the time periods of which were flagged and removed from data analysis. During sampling, three researchers were on board the AML, including a driver, a navigator who also recorded real-time observations of the sampling conditions (e.g., observation of an emissions event), and a researcher dedicated to monitoring the onboard instrumentation.

### 2.3. Data Analysis

Data analysis was conducted using the statistical software R version 3.4 (R Foundation for Statistical Computing, Vienna, Austria). Aromatics measurements (benzene, toluene, and acetaldehyde) exhibited more noise than the other measurements and were smoothed using a 10 second rolling mean prior to analysis. To improve understanding of the contribution of local sources to measured concentrations, the real-time pollutant measurements were used to estimate background concentrations using a method similar to the one described by Brantley et al. [15]. The purpose of the algorithm is to estimate a smooth curve that represents how the minimum of the concentrations varies over time. The algorithm proposed in Brantley et al. [14] involved calculating the minimum values within a given window size (e.g., 10 min) and fitting an ordinary least squares regression spline through the minimums. In this paper, we simplify and improve the algorithm by using a natural spline basis expansion of time with quantile regression rather than choosing a window size and calculating minimums.

Ordinary least squares regression is the most common way of modeling a response *y* as a linear function of a predictor *x*. Given observations $\left(y_{i},\;x_{i}\right)$, where *i* = 1,…,*n*, the intercept, *β*_0_, and slope, *β*_1_ are estimated as the values that minimize
(1)$$\sum_{i=1}^{i=n}{\left(y_{i}-\beta_{0}-\beta_{1}x_{i}\right)}^{2}$$

The resulting estimate of *β*_1_ represents the effect of *x* on the mean of *y.* In cases where the relationship between *y* and *x* is non-linear, a spline basis expansion of *x* can be used to model *y* as a smooth piecewise polynomial function of *x* rather than a simple linear function of *x* [17]. Quantile regression, first proposed by Koenker et al. [18], enables the estimation of the effect of a predictor *x* on a specific quantile of *y* rather than on the mean of *y.* Rather than defining *β*_0_ and *β*_1_ as the solution to the least squares problem (Equation (1)) we will define them as the values that minimize the following objective function: (2)$$\sum_{i=1}^{i=n}\rho_{\tau }(y_{i}-\beta_{0}-\beta_{1}x_{i})$$
$$\rho_{\tau }\left(z\right)=\{\begin{array}{l} \;z\tau \;\mathrm{if}\;z>\;0\\ z\left(\tau -1\right)\;\mathrm{if}\;\;z\leq 0 \end{array}$$
where $\rho_{\tau }\left(z\right)$ is the check loss function, and τ is the quantile level of interest. When the check loss function is used, $\beta_{1}$ represents the effect of *x* on the τ^th^ quantile of *y.* These estimates can be obtained using the quantreg package in R [19].

To estimate the background concentration as a smooth function of time we used quantile regression with a cubic natural spline basis expansion of time with degrees of freedom equal to the number of hours in the time series. The background concentrations were estimated separately for each of the 19 sampling sessions. Several low quantiles were tested, and the results were not sensitive to the level chosen. The results shown are the predicted 10th quantiles of the pollutant concentrations which vary as a smooth function of time (Figure 3). 

To analyze the spatio-temporal trends in the dataset, the sampling route (shown in Figure 1) was divided into 50 m road segments using ESRI ArcGIS. Measurements were then aggregated by taking the median of the observations in each road segment for each of the 5 wind categories, as defined earlier by wind speed and wind direction. Only road segments with at least 5 measurements for a given wind category and within 1 km of the rail yard were used in the analysis. Between 7.1% (benzene) and 11% (BC) of the segments were excluded which is unlikely to influence the results. Medians were used to prevent undue influence by outliers (e.g., emissions very close to the sampling car), based upon previous analysis of mobile monitoring data [15]. 

A total of 2461 aggregated spatial measurements were used in the analyses. While maps and summary statistics provide indication of upwind/downwind differences, regression modeling was conducted to determine whether these differences were statistically significant. Even after aggregation, pollutant distributions were highly skewed and were transformed by taking the logarithm of the measurements in order to meet the normality assumption for the regression analyses. As an initial step to compare the spatial distributions of the pollutants, Pearson correlation coefficients were calculated using pairwise complete observations. The log-transformed aggregated spatial measurements were used in order to reduce the influence of large outliers. 

To determine whether wind conditions or distance from the rail yard resulted in statistically significant differences in mean concentrations, the aggregated pollutant concentrations were modeled as: *Y(s,w) = β*_0_*+ β*_1_*(calm(w)) + β*_2_*(railyard(s)) + β*_3_*(downwind(s,w)) + β*_4_*(distance(s)) + ε(s,w)*(3)
where *Y(s,w)* represents the natural logarithm of the aggregated pollutant concentration at route segment *s* and wind category *w*; *calm(w)* is an indicator of hourly wind speed less than or equal to 0.5 m/s; *railyard(s)* is an indicator variable that takes a value of 1 if route segment *s* is within the railyard and 0 otherwise (Figure 1); *downwind(s,w)* takes a value of 1 if the route segment *s* is downwind of the rail yard for wind category *w*, e.g., wind from the SSW and route segment on the eastern side of the railyard and 0 otherwise; *distance(s)* is the mean distance of the measurements within the route segment to the railyard edge in kilometers, measurements within the rail yard have a distance of 0.

Because the concentrations being modeled were observed across space, they are very likely to be correlated with one another, i.e., concentrations that are close to one another in distance are more likely to be similar than concentrations that are farther apart. The lack of independence in the observed concentrations can result in confidence intervals and *p*-values that are too small when standard OLS is used. To prevent erroneous conclusions based on the assumption of independence, the error term, *ε(s,w)*, was modeled two separate ways. In the “Independent” model the errors were assumed to be independent across both space and wind category (OLS). For this model the parameters were estimated using the lm function in R. In the “Spatial” model, the errors were assumed to be independent across wind category, but correlated across space. Spatial correlation was accounted for by modeling the error term as a Gaussian process with an exponential covariance function. The covariance between the errors at location s_1_, and location s_2_, is defined as a function of the Euclidean distance between the locations (*d*) calculated in meters, a range term (φ), the spatial sill (σ^2^), and the nugget (τ^2^).
Cov(ε(s_1_, t), ε(s_2_, t)) = σ^2^ exp(-φ*d*)(4)
Var(ε(s, t)) = σ^2^ + τ^2^(5)

The parameters in the spatial model were estimated by maximizing the likelihood function using the optim function in R. 

## 3. Results and Discussion

### 3.1. Near-Railyard Pollution Trends

#### 3.1.1. Concentrations and Estimated Background Contribution

For each of the 19 sampling sessions the mean concentration and mean of the estimated background concentrations were determined (Table 2). Although short-term transient events can visually appear to dominate the mean (Figure 3), EXT, CO, acetaldehyde, and PN had substantial background contributions (>60%) attributed to the general Atlanta-area air pollution. At the other end of the spectrum were pollutants more heavily dominated by local emissions, with background contributing less than 30% to BC and oxides of nitrogen. Finally, benzene was approximately split between background and local contributions to the signal. On a session by session basis, there was variability in the background contribution. EXT had the narrowest range of background contribution, likely due to the significant secondary aerosol fraction of the particulate matter. Meanwhile, other parameters had wider ranges, such as the background BC contributing from 7%–53%.

#### 3.1.2. Spatial Variation within Route under Various Wind Conditions

After combining multiple mobile monitoring sessions based on the wind category, significant variability in 50 m median concentrations is observable along the mobile monitoring route and also between upwind/downwind areas (Figure 4). For mapping, concentrations were divided into 8 bins with equal numbers of road segments assigned to each color category to highlight spatial differences. Maps of the other pollutants are included in Appendix A. Visual inspection of maps of pollutant concentrations under different wind conditions indicated that NO_2_ concentrations in areas close to the railyard were elevated under downwind conditions (Figure 4) and concentrations measured within the railyard were generally higher than concentrations on nearby roads with low traffic. Some of the downwind areas close to the railyard had comparable concentrations to those observed on a higher traffic road sampled during some of the driving routes. The section of the driving route passing within the two railyards had consistently higher NO_2_ levels with respect to upwind areas or low traffic roadways. Similar trends were observed in the NO and NO_y_ concentrations as well as PN and BC (Appendix A). These similarities are unsurprising, given diesel combustion typically co-emits NOx, BC, and large numbers of particles.

The classification of the measurements into upwind, downwind, and within-yard during different wind conditions provides information on a near-field pollution effect, or lack thereof. In Figure 5, geometric means represent the mean of the log concentrations transformed back into the original units. This is the response that is modeled in the regression analysis to determine if any of the predictors have a significant effect on concentrations. NO_2_ and PN have an apparent increase in their geometric mean in areas downwind of the railyard, i.e., the East side when the wind is from the SSW and the West side when the wind is from the NNE, as well as within the yard. BC and EXT—an indicative value of total particulate matter—are also elevated within the rail yard with some evidence of a downwind effect. Meanwhile, benzene and CO exhibit only slight to no change between zones during the two crosswind conditions.

In absolute terms, concentration levels are similar, or lower, to those observed in past near-railyard and near-road studies in the United States. Prior to the mobile monitoring study, Galvis et al. [9] reported slightly higher average BC concentrations, with mean values of 1.3–1.5 µg m^−3^, however these values were measured over a full year. A monitoring site situated adjacent to a major railyard in Cicero, IL measured overall mean values of BC and NO_2_ levels of 0.64 µg m^−3^ and 20.9 ppb, while mean values during timeframes downwind of the railyard were 0.82 µg m^−3^ and 27.3 ppb, respectively, over a ~6 month period of time [10].

#### 3.1.3. Multipollutant Spatial Correlation

Pearson correlation coefficients calculated using the 50 m spatial medians revealed the most highly correlated pollutant measurements were NO_y_ with NO_2_ and NO with correlations of 0.90, 0.80 respectively (Figure 6). Moderate correlation was observed between BC, PN, and oxides of nitrogen. EXT had moderately low correlation with BC, PN, and oxides of nitrogen, likely due to a significant portion of the EXT signal being regional in nature. Toluene was moderately correlated with both benzene and CO with correlations of 0.59 and 0.51, respectively. Overall, these correlation findings, along with the upwind-downwind analyses, indicate that NOx, BC and PN are co-emitted and have similar trends in areas surrounding the railyard, whereas other measured species were likely emitted by separate sources not strongly correlated to each other. Past in-depth analysis of mobile monitoring data processing methods by Brantley et al. [15] suggests that these results may vary based upon whether the mobile monitoring data had been evaluated in a raw, very high time resolution form, smoothed temporally, versus aggregated into spatial segments. Spatial aggregation into median values was determined by Brantley et al. [15] to limit the influence of extreme outliers. Additionally, past analysis revealed that spatial rather than temporal data aggregation appeared to most clearly isolate geographic areas with higher versus lower observed concentration [15].

### 3.2. Effect of Meteorology and Distance to Railyard 

Regression analysis isolated the effects associated with multiple potentially influential factors on the transformed concentrations (Table 3). The intercept estimate, *β*_0_, represents the predicted mean of the aggregated log-transformed concentrations at the edge of the railyard under upwind and parallel wind conditions. Consequently exp(*β*_0_), represents the predicted mean of the aggregated concentrations under these conditions in the original measurement units. Similarly, exp(*β*_1_) represents the factor by which the aggregated concentrations change under calm or variable wind conditions compared to the intercept, and exp(*β*_4_) represents the factor by which the aggregated concentrations change with each kilometer increase in distance from the edge of the rail yard. An estimated value of *β*_4_ that is significantly less than 0, (exp(*β*_4_) significantly less than 1) represents an exponential decay in concentration with distance from the rail yard with a rate corresponding to the estimated value of *β*_4_.

Calm or variable wind conditions significantly increased the concentrations of all the pollutants measured, with the greatest effect on the nitric oxides which increased by a factor of 1.5 to 2 depending on the pollutant and model, and the smallest effect on CO, benzene and acetaldehyde which increased by factors of 1.1 to 1.2 (Appendix A). Near-source studies commonly focus on time periods of cross-wind conditions to clearly isolate spatial gradients in pollution attributed to the source. However, low speed, mixed winds during calm periods can limit the dispersion of emissions in the immediate vicinity of the rail yard as well as in the surrounding urban area, resulting in higher concentrations.

For both regression analyses, being within the rail yard, or downwind of the rail yard has a significant positive effect on nitrogen oxides, BC, and EXT. These pollutants, along with PN also exhibit a significant negative correlation with distance from the rail yard. NO displayed the largest relative decrease with distance, decreasing by a factor of approximately 0.5 with a distance of 1 km, while BC and PN both decreased by a factor of approximately 0.7. In addition to dispersion of emissions with distance, another effect on NO is the post-emissions conversion to NO_2_. EXT, which was observed to have a larger fraction of the mean concentration attributable to background decreased by a factor of approximately 0.8. 

The standard errors of the estimates are generally larger for the model with spatially correlated errors. This is a more accurate representation of the uncertainty in our estimates of the parameters, because measured concentrations on nearby road segments are unlikely to be independent. However, in the majority of cases the conclusions based on the independent model do not differ from those of the spatial model.

## 4. Conclusions

This study utilized a mobile air monitoring vehicle outfitted with advanced air measurement instrumentation to map air pollution levels surrounding a major railyard area in Atlanta, GA. Beyond the specific findings at this site, this study more broadly demonstrates how a short-term intensive series of measurements on a mobile platform can yield a rich set of data characterizing the spatial variability of air pollution surround a large area source. As discussed in a recent study applying mobile monitoring near port terminals [20], the suitability of mobile monitoring as an approach for near-source assessment depends upon roadway access to areas near the source and a minimization of other large nearby sources that may confound data interpretation. While the Atlanta railyard area studied was suitable for mobile monitoring, one limitation to note is the inability to distinguish impacts between the two conjoined railyards. Beyond mobile monitoring study design, local-scale air quality trends were discerned using data processing, aggregation, and statistical models to account for background pollution levels, bin multiple runs by wind condition, and quantify the influence of multiple parameters on spatial patterns of pollutants. 

In this study, elevated near-field air pollution was evident for a number of pollutants (BC, NO/NO_2_/NO_y_, PN, EXT), with an apparent inverse relationship with distance from the railyard boundary. Low speed wind conditions were associated with increases of all measured pollutants, including air toxics and CO which did not have any clear relationship with proximity to the railyard. The low wind speed conditions confound the clear discernment of upwind/downwind areas and attribution of any portion of the observed pollution to the local source. 

Conducting mobile monitoring with advanced air monitoring instrumentation or installing reference-level analyzers in near-source areas are costly endeavors, which has to date limited the number of near-railyard field studies conducted. Emerging air sensor technologies are expected to vastly increase the amount of near-source observational data in the future, for a limited number of pollutant types [21]. This present study indicates that specific air pollution measurements (e.g., BC, NO) that are directly emitted are strong indicators of the near-field emissions impacts on air quality. 

As populations and transportation sources naturally concentrate in urban areas, assessing near-field impacts is anticipated to be of growing interest. This study displayed that concentration gradients are evident in near-railyard areas and pollutants characteristic of diesel emissions co-varied. Given the variety in railyard designs and surrounding environments, similar field studies conducted in other areas would broaden the body of information available on near-field air pollution impacts.

## Figures and Tables

**Figure 1 ijerph-16-00535-f001:**
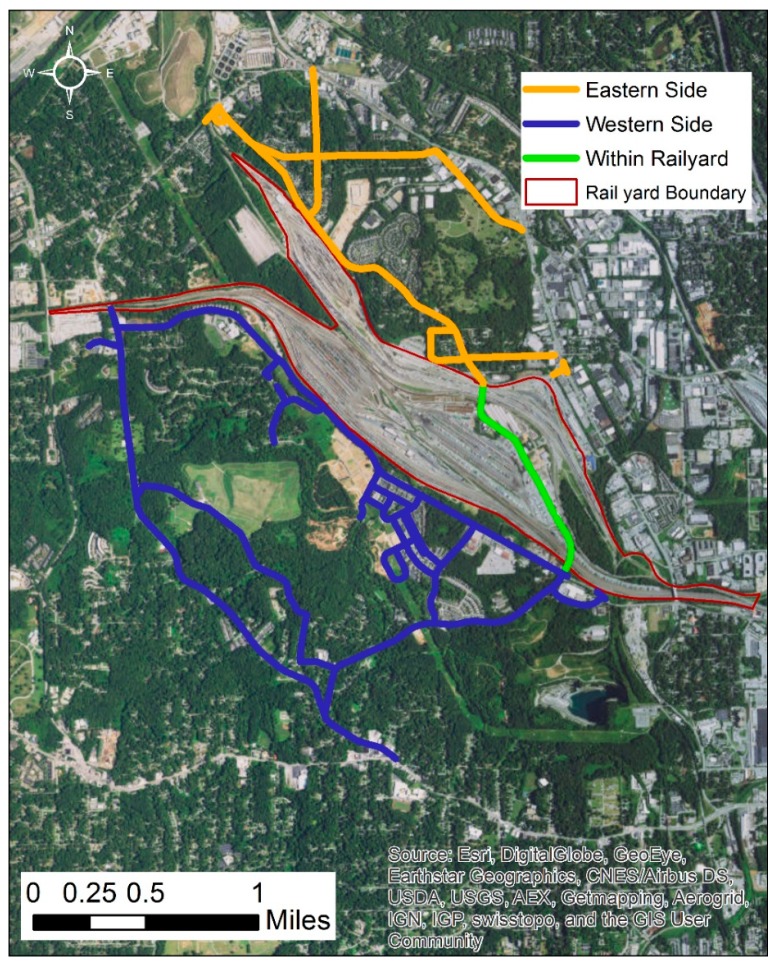
Map of conjoined Inman/Tilford rail yard areas with locations surveyed in detail by the mobile monitoring laboratory shown in blue (western side), green (crossing through the rail yard), and eastern side (orange).

**Figure 2 ijerph-16-00535-f002:**
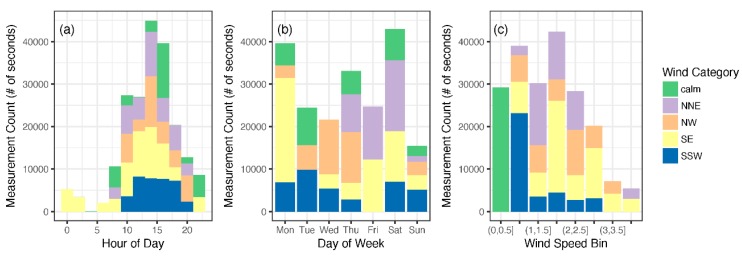
Measurement count by (**a**) time-of-day and wind category (**b**) day-of-week and wind category and (**c**) wind speed and wind category.

**Figure 3 ijerph-16-00535-f003:**
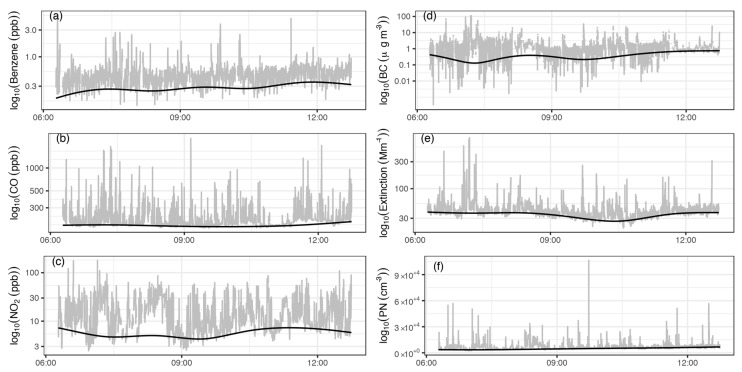
Example sampling session (concentration values in black) and estimated background concentrations (shown in black).

**Figure 4 ijerph-16-00535-f004:**
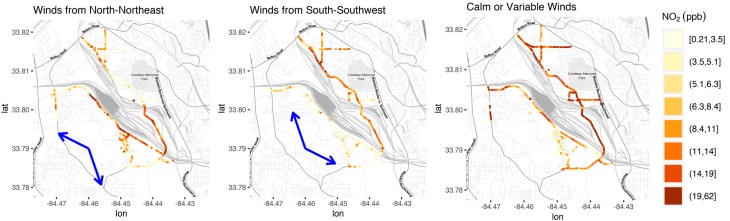
Concentrations of NO_2_ aggregated by 50 m road segment during (**left**) winds from the north-northeast (**middle**) winds from the south-southwest and (**right**) calm winds. Segments are binned so that each color represents an equal number of road segments. Blue arrows represent range of mean hourly wind directions.

**Figure 5 ijerph-16-00535-f005:**
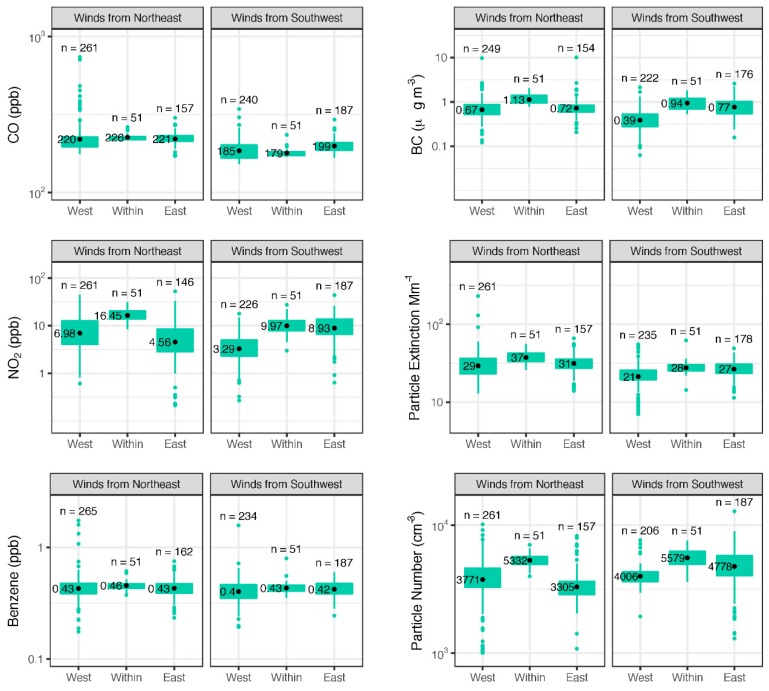
Concentration geometric means (black points) and box-and-whisker plots for concentrations aggregated by 50 m road segment and hour on road segments representing upwind, within, and downwind of the rail yard during cross-wind conditions. Boxes represent 25th and 75th quantiles, outlier points are measurements greater than 1.5 times the interquartile range away from the edge of the box. All measurements are plotted on a log-scale.

**Figure 6 ijerph-16-00535-f006:**
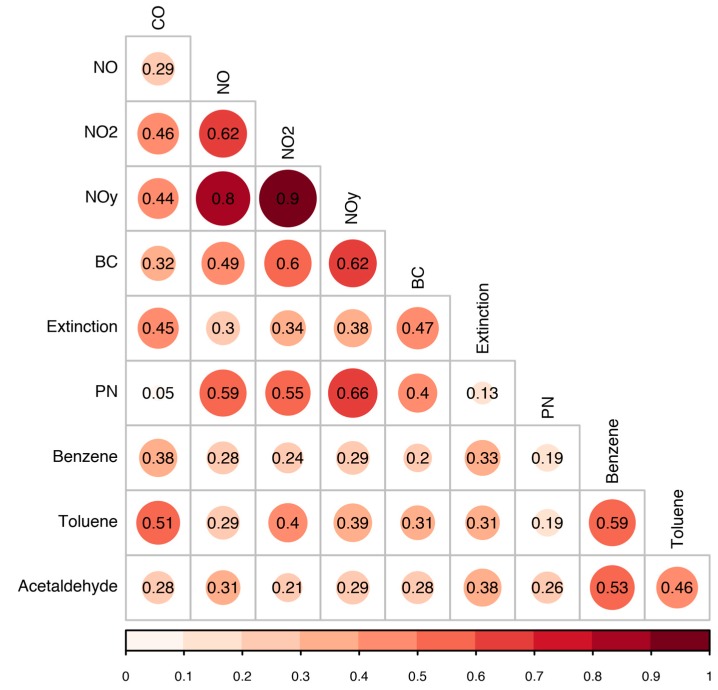
Pearson correlation between pollutant concentrations aggregated by 50 m road segment and hour.

**Table 1 ijerph-16-00535-t001:** Mobile monitoring instrumentation and sampling rates.

Measurement	Rate	Instrument
Carbon Dioxide (CO_2_)	0.9 s	Licor 6262 (2) and Licor 820
Carbon Monoxide (CO)	1 s	Aerodyne mini QC-TILDAS ^1^ (2230 cm^−1^)
Nitric Oxide (NO)	1 s	Thermo 42i Chemiluminescence
Nitrogen Dioxide (NO_2_)	1 s	Aerodyne Cavity Enhanced Phase Shift
Oxides of Nitrogen (NOy)	1.4 s	Thermo 42i with external inlet-tip Mo Converter
Black Carbon (< 2.5 µm)	3 s	Thermo 5012 Multi-Angle Absorption Photometer
Particle Extinction	3 s	Aerodyne Cavity Enhanced Phase Shift
Particle Number Concentration	1.8 s	TSI 3025A Condensation Particle Counter
Aromatics and Oxygenates, including benzene, toluene, acetone, acetaldehyde	1.4 s	Ionicon Quadrupole PTR-MS ^1^

^1^ QC-TILDAS: Quantum Cascade Tunable Infrared Laser Differential Absorption Spectrometer; PTR-MS: Proton Transfer Reaction Mass Spectrometer.

**Table 2 ijerph-16-00535-t002:** Summary of measured pollutant concentrations by session. Column “N” represents number of sessions while N (obs) is number of observations. The grand means of the observed concentrations, estimated background concentrations, and proportion of the mean concentration attributed to background are given in the last three columns. Numbers in parentheses represent the standard deviations of the session means.

Pollutant	N	N (obs)	Session Mean	Background Mean	Background Fraction
BC (μg m^−3^)	18	157107	1.4 (0.64)	0.33 (0.16)	0.27 (0.14)
CO (ppb)	19	183693	270 (72)	190 (24)	0.74 (0.12)
PN (cm^−3^)	18	173049	6300 (2300)	3900 (1600)	0.61 (0.11)
Ext. (Mm^−1^)	15	150828	35 (13)	27 (11)	0.76 (0.077)
NO (ppb)	19	183041	11 (7)	1 (0.59)	0.1 (0.043)
NO_2_ (ppb)	19	181870	16 (7.3)	5.2 (5)	0.28 (0.14)
NO_y_ (ppb)	18	181120	23 (12)	5.5 (2.5)	0.25 (0.068)
Acetaldehyde (ppb)	17	189208	2.2 (0.36)	1.5 (0.3)	0.68 (0.075)
Benzene (ppb)	17	189208	0.51 (0.073)	0.26 (0.057)	0.51 (0.095)
Toluene (ppb)	17	189208	0.48 (0.18)	0.13 (0.032)	0.28 (0.081)

**Table 3 ijerph-16-00535-t003:** Regression Results. Estimated effect of each of the predictors on the mean log transformed pollutant concentration, using both the independent and spatially correlated error models. Standard errors of the estimates are shown in parenthesis. Estimates that are significantly different from 0 (*p*-value less than 0.05) are marked with *.

	*β* _0_ *(intercept)*		*β* _1_ *(calm)*		*β* _2_ *(railyard)*		*β* _3_ *(downwind)*		*β* _4_ *(distance)*	
	Independent	Spatial	Independent	Spatial	Independent	Spatial	Independent	Spatial	Independent	Spatial
NO	0.76 (0.02) *	0.66 (0.09) *	0.41 (0.03) *	0.53 (0.19) *	0.66 (0.05) *	0.59 (0.08) *	0.11 (0.04) *	0.37 (0.09) *	−0.90 (0.06) *	−0.79 (0.08) *
NO_2_	2.06 (0.03) *	1.85 (0.1) *	0.46 (0.04) *	0.71 (0.22) *	0.62 (0.05) *	0.67 (0.08) *	0.21 (0.04) *	0.51 (0.09) *	−0.75 (0.07) *	−0.47 (0.09) *
NO_y_	2.27 (0.02) *	2.16 (0.07) *	0.39 (0.03) *	0.54 (0.15) *	0.59 (0.04) *	0.58 (0.06) *	0.17 (0.03) *	0.34 (0.07) *	−0.73 (0.05) *	−0.54 (0.07) *
BC	−0.43 (0.02) *	−0.50 (0.06) *	0.35 (0.03) *	0.38 (0.13) *	0.58 (0.04) *	0.59 (0.05) *	0.20 (0.03) *	0.38 (0.05) *	−0.4 (0.05) *	−0.32 (0.06) *
CO	5.31 (0.01) *	5.30 (0.02) *	0.12 (0.01) *	0.19 (0.05) *	0.02 (0.01)	0.04 (0.02) *	0.03 (0.01) *	0.02 (0.02)	0.05 (0.01) *	0.09 (0.02) *
Ext.	3.30 (0.01) *	3.25 (0.06) *	0.34 (0.02) *	0.40 (0.13) *	0.21 (0.03) *	0.24 (0.04) *	0.09 (0.02) *	0.16 (0.04) *	−0.18 (0.04) *	−0.22 (0.04) *
PN	8.42 (0.01) *	8.38 (0.04) *	0.15 (0.02) *	0.19 (0.08) *	0.27 (0.02) *	0.24 (0.03) *	0.002 (0.02)	0.13 (0.04) *	−0.30 (0.03) *	−0.30 (0.04) *
Benz.	−0.85 (0.01) *	−0.84 (0.02) *	0.13 (0.01) *	0.14 (0.03) *	0.03 (0.02) *	0.01 (0.03)	0.01 (0.01)	−0.01 (0.03)	−0.06 (0.02) *	−0.07 (0.04)
Tol.	−1.31 (0.01) *	−1.23 (0.05) *	0.30 (0.02) *	0.32 (0.07) *	0.24 (0.03) *	0.12 (0.06) *	−0.01 (0.02)	−0.10 (0.07)	0.12 (0.04) *	0.04 (0.09)
Acetal.	0.70 (<0.01) *	0.73 (0.02) *	0.13 (0.01) *	0.13 (0.03) *	0.10 (0.01) *	0.03 (0.02)	0.08 (0.01) *	0.05 (0.02) *	−0.06 (0.01) *	−0.14 (0.03) *

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
