# Peer review of "Characterization of Spatial Air Pollution Patterns Near a Large Railyard Area in Atlanta, Georgia"

_ijerph, 2019, doi:10.3390/ijerph16040535_

Round 1
Reviewer 1 Report
Brantley et al describe mobile measurements of air pollutant concentrations in the vicinity of a railyard in Atlanta. The manuscript is written well and easy to follow.
My main criticism is that the manuscript reads more as a "presentation of results" than a discussion of the novel aspects of the data collection or analysis. Presumably since this is a case study of a single railyard in Atlanta, the specific concentrations measured in this study are of less general interest than methodological or analytical insights (e.g., how to use mobile data to investigate large area sources).
Pollutant concentrations are generally higher on the railyard property or downwind of the railyard, which is probably to be expected. I think the paper would be more impactful if the authors focused more on what new information was learned or on how this study may translate to future studies. For example, there is lots of interesting structure to the data shown in Fig 3 that is not discussed much in the text. All of the measured species display peaks, but peaks of different species are not all coincident (for example, peaks of PN and BC do not seem to be strongly correlated). Some investigation of multi-pollutant patterns and correlation is covered in the Pearson R matrix (Fig 6), but that is highly aggregated. A potentially more informative analysis would be to see if emission factors can be determined (e.g., BC/CO ratios).
Comments
The panels in Figure 3 are very small, and it's hard to discern the baseline from the full data. In some cases it looks like the baseline is higher than the measurement. It would help to make this figure have one larger panel, or zoom in on a portion of the data, and put the rest of the panels in the SI.
The "width" of the baseline fitting (lines 145-163, Figure 3) is unclear. Is the regression completed for each sampling day, or hourly, etc?
Figure 4 - the legends are hard to read. The figure needs larger legends or higher resolution.
Lines 230-240 note that for some species the background contributed ~60% of the signal, but for others the background contributed less. However, since figure 3 is zoomed out so far, this distinction is hard to tell visually.
Line 281-282 - what is the difference between the numbers in the main text and in parenthesis?
Section 3.1.3 - are the correlations calculated for the total concentration or the background-adjusted concentration?
The authors should place these results in better context of human exposures. For example, are the concentration increases downwind of the railyard important from a human health perspective? More broadly, how important are railyards as a source of human exposures to ambient air pollution? Are they frequently located in residential areas (the area south of this railyard seems to be residential) such that large populations could be exposed?
What is the difference between "independent" and "spatial" in Table 3? I think this is what lines 3003-307 describe, but it is not clear. This distinction in the regression models does not seem to be described in the Methods section.
The regression models in Table 3 do not seem to ad much value, nor are the results discussed in much detail. The regression models seem to fit the expected structure (e.g., higher concentrations near the railyard), but do they teach us anything new about this source? How could these models be used in planning or policy making or interpreting results?
Minor/grammatical comments
Line 37 - the reference to the HEI 2010 report should have the same format as all of the other references (i.e., numerical).
Line 66 "Cahill et al"
Reviewer 2 Report
1. Figure 3: Plot such that Y axis is log scale. The point of the figure is to show how well the methodology captured the background and current plotting scheme compresses the data such that the baseline trend cannot be read.
2. Add wind rose to Figure 4. Is the satellite image background helping with the visual clarity? The data dots are rather small and pattern is obscured by all the background color. Would a simpler depiction in ARCGIS (say, gray background with railyard features plotted) be clearer? In the similar figures in the Supplement, add wind rose or a simple arrow to indicate wind directions. It will make for a much quicker interpretation.
3. Table 2: if NO and NO2 sampling rates differ by a factor of 5, why are the number of observations comparable? Also, see comment 5 below.
4. Line 232: What definition or thresholds were used to call the concentrations “low to moderate?” refrain from subjective assessments.
5. Line 237-238: Quantify the variability. Add stats to table 2. Range is good to know, but please add std. dev if the authors think parametric stars are appropriate for a rather small sample size of 15-19. Medians and quantiles would be much preferable in the right three columns in this Table. I see a switch to non-parametric stats later in the paper.
6. It is difficult to find novel conclusions that can be drawn based on the results of this study in the Conclusions section. For example, in the first paragraph it is only the very last sentence that draws on the results of this study. So why not begin with that than the general background which is discussion material? Third and fourth paragraphs are discussion. Revise the content or rename the section.
7. A summary of content in lines 332-337 belongs in the abstract – clear useful result that is a good takeaway for the readers. Also revise abstract so it is sparser on the details of the methods. The last line of the abstract is not really a novel finding specific to this study so perhaps that is not abstract material?
Reviewer 3 Report
SummaryThis paper describes measurements of air pollutants (black carbon, NO, NO2, NOy, particle number, and CO) plus certain organic compounds in and near a railyard. It represents an exploration of air pollutant levels and factors affecting those levels in an understudied near-source environment.
Minor Revisions
Line 20: The abstract should state that measurements were made in May 2012. Additional information could include how many hours were in each session and that the measurements were conducted over different days of the week and wind conditions.
Line 20-21: How are “low to moderate concentrations” defined?
Line 25: “Wind category” could be defined as wind direction relative to the railyard to clarify that wind speed was not considered.
Lines 25-27: Were correlations low for all other pairs of measured pollutants.
Line 89: How long was each monitoring session? The methods should also state how the routes were selected, how fast the mobile monitoring vehicle was driven, and whether driving was continuous or stops were made along the route.
Figure 1: If possible, consider adding the sites of the meteorological stations to the study area map.
Lines 108-110: The description of the wind direction categories is confusing because the sectors are not all the same size. It may be helpful to describe how the sectors were chosen to have small sectors parallel to the roads and large upwind/downwind sectors.
Figure 2: I like this figure! Have you considered complementing it with a wind rose?
Table 1: This table should list the model and manufacturer of each instrument. For example, PNC results can depend on which condensation particle counter is used.
Line 141: Which version of R was used? Are there any other packages that should be listed here?
Line 145: The “method similar to the one described by Brantley et al. [12]” should be briefly described here.
Lines 148-226: The description of the data analysis is quite statistically involved. Can the methods be explained more generally in plain English for a less statistically sophisticated audience?
Line 152: Which predictors were considered for the models, and how were they selected? Were other factors like temperature, time of day, or day of week also important?
Line 160: What is z in equation 2?
Line 175: Specify that these are wind direction categories.
Lines 176-177: How much of the data was excluded, and would this be likely to influence the results?
Lines 180-189: This section appears to describe results that are out of place in the methods section.
Lines 241-244: The Table 2 caption is very long and difficult to parse. Consider splitting it into smaller sentences.
Line 265: The text describes “areas downwind of the railyard” while the figures compare areas east and west of the railyard. Both should use the same metric of location relative to the railyard.
Line 276: Add to the Figure 5 caption that points represent geometric means.
Lines 277-282: The measured concentrations from this study could be mentioned here for comparison with the other studies because the reader may have forgotten the measured concentrations by this point of the paper.
Figure 6: It is not necessary to show that pollutants have a correlation of 1 with themselves.
Line 304: It would be helpful to discuss which variables are important before describing uncertainty in the models.
Line 316: Delete the extra instance of “shown in parentheses” because as written it seems to say that the effect estimates are in parentheses.
Lines 333-337: How were the factor decreases obtained?
Lines 329-330: Delete the redundant phrase “which was the section of the route when the mobile monitoring vehicle traveled on a public roadway passing through the rail yard (Fig. 1).
Lines 343-345: The paper by Arunachalam would be good to also mention in the introduction.
Supplemental information: The figure and table captions should be more detailed, with a similar level of detail to the captions in the main text. Do the correlations change when you consider only measurements made on the same side of the railyard?
Typographic Revisions
Line 37: HEI (2010) is not in the reference list.
Line 88: The mobile monitoring vehicle has not yet been introduced at this point of the paper. Consider replacing “The … vehicle” with “A … vehicle.”
Line 104: Delete “Galvis et al.”
Line 132: The sentence should end with a period instead of a comma.
Line 155: Was reference 14 (not 15) meant here?
Line 163: Was reference 15 (not 16) meant here?
Line 262: Delete “are labeled because they.”
Line 319: Delete the extra period at the end of the Table 3 caption.
Round 2
Reviewer 3 Report
I would like to thank the authors for their careful responses to my earlier comments.